# Calculation of the Overlap Function and Associated Error of an Elastic Lidar or a Ceilometer: Cross-Comparison with a Cooperative Overlap-Corrected System

**DOI:** 10.3390/s20216312

**Published:** 2020-11-05

**Authors:** Michaël Sicard, Alejandro Rodríguez-Gómez, Adolfo Comerón, Constantino Muñoz-Porcar

**Affiliations:** 1CommSensLab, Department of Signal Theory and Communications, Universitat Politècnica de Catalunya (UPC), 08034 Barcelona, Spain; alejandro@tsc.upc.edu (A.R.-G.); comeron@tsc.upc.edu (A.C.); constan@tsc.upc.edu (C.M.-P.); 2Ciències i Tecnologies de l’Espai-Centre de Recerca de l’Aeronàutica i de l’Espai/Institut d’Estudis Espacials de Catalunya (CTE-CRAE/IEEC), Universitat Politècnica de Catalunya (UPC), 08034 Barcelona, Spain

**Keywords:** elastic lidar, overlap function, cross-comparison, error estimation

## Abstract

This paper establishes the relationship between the signal of a lidar system corrected for the incomplete overlap effect and the signal of another lidar system or a ceilometer for which the overlap function is unknown. Simple mathematical relationships permit the estimation of the overlap function of the second system as well as the associated error. Several overlap functions have been retrieved with this method over a period of 1.5 years with two lidar systems of the Universitat Politècnica de Catalunya, Barcelona, Spain. The error when the overlap function reaches 1 is usually less than 7%. The temporal variability estimated over a period of 1.5 years is less than 11% in the first 1.5 km from the surface and peaks at 18% at heights between 1.7 and 2.4 km. The use of a non-appropriate overlap function in the retrieval of the backscatter coefficient yield errors up to 60% in the first 0.5 km and up to 20% above.

## 1. Introduction

The overlap function between the laser beam and the receiver field of view of a lidar system, henceforth referred to as simply the “overlap function”, of an aerosol lidar system or a ceilometer quantifies in a range-resolved manner the number of photons arriving at the detector normalized to the number of backscattered photons arriving at the telescope aperture. It is a function of the optical/geometric arrangement of the transmitting and receiving optics of the system. In particular, it varies with distance and depends on the laser beam diameter, shape, and divergence, on the receiver properties (telescope diameter, equivalent focal length, field stop diameter), and on the location of the emitter and receiver optical axes relative to each other [1,2]. In general, many of the rays arriving at the telescope aperture from atmosphere volumes illuminated by the laser beam at short distances cannot pass through the field stop and reach the detector; thus, the overlap function usually starts at zero at the lidar/ceilometer from where it starts to increase. In aerosol application, the optical/geometric arrangement is usually set up so that the overlap function at long distances reaches unity (full overlap). Many works have assessed the overlap function of different systems either analytically (e.g., [3]) or through ray tracing (e.g., [4,5]). In practice, an accurate theoretical calculation of the overlap function based on optical parameters alone is not practical, because most of these are difficult to measure or estimate [6]. Thus, in many cases, the overlap function has to be estimated experimentally.

Low-energy lidar systems and ceilometers designed for tropospheric studies must have a very narrow field of view in order to suppress the background radiation and maximize the signal-to-noise ratio, especially above the planetary boundary layer (PBL). The main drawback of this technical requirement is that it brings the distance of full overlap at large distances, usually way beyond the PBL. Then, the overlap correction becomes a crucial and mandatory issue. With the recent deployment of low-energy lidars and ceilometers in national and international networks, the overlap correction represents a key issue affecting directly the quality of the data. In addition, time- and temperature-dependent mountings can alter the overlap function without warning. Automatic and continuous estimation is desired to monitor the overlap function. Note that some ceilometers, especially designed for planetary boundary layer studies, do not have this drawback because they are designed with a broader field of view so as to maximize the signal-to-noise ratio in the planetary boundary layer and lower the incomplete overlap effect as much as possible.

Despite the necessity, it seems that the development of techniques to estimate and monitor the overlap function is lengthy and has not reached a consensual point of agreement worldwide. There are a few well-known solutions used by the lidar community: estimation in a homogeneous atmosphere [7] and the use of a cooperative Raman channel [8], to name but a few. The practicality of the first technique depends on the state of the atmosphere and thus represents a limiting factor for regular monitoring. The second one requires a Raman channel that often low-energy lidars and ceilometers do not have. More recently, a novel technique is being developed for the Micro Pulse Lidar (MPL) systems, a number of which are in active use as part of NASA’s Micro Pulse Lidar NETwork (MPLNET [9]). It consists of adding a secondary wide field of view receiver to each MPL system [10,11]. The technique is being progressively deployed at the key sites of the network [9]. To get rid of the overlap issues, some systems duplicate their number of signals having both far and near range receivers. The near-range receivers usually allow lowering the height of full overlap down to ≈100 m. This is the case of the Polly^XT^ systems [12].

In this paper, we propose a method to determine experimentally the overlap function of a low-energy lidar or a ceilometer by cross-comparing its signal to that of a cooperative overlap-corrected, reference lidar system. No technical modification is required on the lidar/ceilometer, the overlap function of which has to be estimated. Although the method is conceptually similar to other studies (see e.g., [13]), our work includes the analytical formulation of the error associated to the estimated overlap function as a function of the error of the input overlap-corrected signal of the reference system.

## 2. Materials and Methods

Let us assume two lidar systems: System 1 and System 2. System 1 is the reference instrument, the overlap function of which is known. System 2 is an elastic lidar or a ceilometer, and it has an unknown overlap. The method consists of estimating the overlap function of System 2 with its associated error. The assumption is made that both systems work at the same wavelength.

### 2.1. Range-Square Corrected Signal and Associated Error of the Reference System

Let us call $X_{1}^{cor}$ the range-square corrected signal of System 1 corrected for its overlap. At a given distance z, it can be expressed as a function of the atmospheric properties as:(1)$$X_{1}^{cor}(z)+\Delta X_{1}^{cor}(z)=C_{1}\beta (z)T^{2}(z)$$
where $C_{1}$ is the system constant, β is the total backscatter coefficient, T is the total transmission and $\Delta X_{1}^{cor}$ is the error associated to $X_{1}^{cor}$. Next, we express $X_{1}^{cor}$ and $X_{1}^{cor}$ independently.

Let us first calculate $X_{1}^{cor}$. The overlap correction of the system taken as reference assumes a previously computed overlap function $O_{1}\left(z\right)$ with its associated error $\Delta O_{1}\left(z\right)$. $O_{1}\left(z\right)$ and $\Delta O_{1}\left(z\right)$ can be determined from the expressions given in Appendix A for a combined elastic–Raman system, or with any other valid and reliable method. From the original, raw, received power $P_{1}\left(z\right)$, we can write:(2)$$X_{1}^{cor}\left(z\right)=z^{2}\cdot \frac{P_{1}\left(z\right)}{O_{1}\left(z\right)}.$$

$\Delta X_{1}^{cor}$ is calculated, taking into account the noise and the error of the overlap function with which the signal has been corrected. By using propagation of errors, we calculate:(3)$$\Delta X_{1}^{cor}\left(z\right)=\sqrt{\frac{z^{4}}{{\left[O_{1}\left(z\right)\right]}^{2}}{\left|\Delta P_{1}\left(z\right)\right|}^{2}+{\left\{\frac{z^{2}\cdot P_{1}\left(z\right)}{{\left[O_{1}\left(z\right)\right]}^{2}}\right\}}^{2}{\left|\Delta O_{1}\left(z\right)\right|}^{2}}$$
where ${\left|\Delta P_{1}\left(z\right)\right|}^{2}$ is the variance of the signal $P_{1}\left(z\right)$.

### 2.2. Overlap Function and Associated Error of System 2

Let us now call $X_{2}^{ncor}(z)$ the range-square corrected signal of System 2, the overlap function of which is unknown and thus not corrected. At a given distance z, $X_{2}^{ncor}(z)$ relates to the atmospheric properties as follows:(4)$$\frac{X_{2}^{ncor}(z)+\Delta X_{2}^{ncor}(z)}{O_{2}(z)+\Delta O_{2}(z)}=C_{2}\beta (z)T^{2}(z)$$
where $C_{2}$ is the system constant, $O_{2}(z)$ is its overlap function, and $\Delta O_{2}(z)$ is the error associated to the retrieval of $O_{2}(z)$. $\Delta X_{2}^{ncor}(z)$ is the noise associated to $X_{2}^{ncor}(z)$ and estimated as the variance of the original, raw, received power $P_{2}\left(z\right)$ multiplied by $z^{2}$. Next, we combine Equations (1) and (4) to solve for $O_{2}(z)$ and $\Delta O_{2}(z)$.

If we call Norm the ratio of $C_{2}$ to $C_{1}$, which can be obtained by dividing the two corresponding signals at high altitude where the overlap function of both systems can be assumed unity, we obtain:(5)$$\frac{X_{2}^{ncor}(z)+\Delta X_{2}^{ncor}(z)}{O_{2}(z)+\Delta O_{2}(z)}=\left(X_{1}^{cor}(z)+\Delta X_{1}^{cor}(z)\right)Norm.$$

By re-arranging terms, we finally can write:(6)$$O_{2}(z)+\Delta O_{2}(z)=\frac{1}{Norm}\frac{X_{2}^{ncor}(z)}{X_{1}^{cor}(z)}\left[1+\frac{1}{X_{2}^{ncor}(z)}\frac{X_{1}^{cor}(z)\Delta X_{2}^{ncor}(z)-X_{2}^{ncor}(z)\Delta X_{1}^{cor}(z)}{X_{1}^{cor}(z)+\Delta X_{1}^{cor}(z)}\right]$$
from which we can retrieve $O_{2}(z)$ as:(7)$$O_{2}(z)=\frac{1}{Norm}\frac{X_{2}^{ncor}(z)}{X_{1}^{cor}(z)}$$
and $\Delta O_{2}(z)$ as:(8)$$\Delta O_{2}(z)=\frac{1}{Norm}\frac{\Delta X_{2}^{ncor}(z)-\frac{X_{2}^{ncor}(z)}{X_{1}^{cor}(z)}\Delta X_{1}^{cor}(z)}{X_{1}^{cor}(z)+\Delta X_{1}^{cor}(z)}.$$

$O_{2}(z)$ is constrained to unity for $z\geq z_{0}$. $z_{0}$ is a distance chosen through visual inspection where both range-square corrected signals have been overlapping for at least a few hundred meters.

## 3. Results

To illustrate the method, we use two lidar systems. System 1 and System 2 are the UPC/EARLINET (Universitat Politècnica de Catalunya/European Aerosol Research Lidar Network, Barcelona, Spain) multi-wavelength system and the UPC/MPLNET micro pulse lidar, respectively. System 1 is part of the EARLINET/ACTRIS (Aerosols, Clouds and Trace gases Research InfraStructure Network; https://www.actris.eu/default.aspx) network, while System 2 is part of MPLNET (https://mplnet.gsfc.nasa.gov/). System 1 is currently an eight-wavelength system with three elastic, three Raman (2 N_2_ and one water vapor), and two depolarization channels. It is described in [14] and [15]. The overlap function of System 1 is regularly estimated to satisfy the ACTRIS quality assurance/quality control procedures. The method employed for the overlap estimation is the one [8] applied to nighttime measurements. The analytical formulation for the calculation of $O_{1}\left(z\right)$ and $\Delta O_{1}\left(z\right)$ is given in Appendix A. System 2 is the polarized MPL described in [16]. The system achieves complete overlap between 4 and 6 km according to [6,10]. These limits can adjust up and down due to thermal–mechanical changes in the system.

We use the wavelength of 532 nm, which both systems have. System 1 has a vertical resolution of 3.75 m, and System 2 has a vertical resolution of 15 m. To compare profiles from both systems, the resolution of System 1 has been downgraded to that of System 2. All measurements presented are nighttime measurements of a 150-min duration, and they were recorded simultaneously. Both systems are 30 m apart. Time and space co-location guarantees that data are under the same atmospheric conditions. The use of the Raman method to retrieve $O_{1}\left(z\right)$ forces the analysis to be performed only with nighttime measurements. The four selected nighttime measurements were performed just after a realignment (a couple of hours at most) of both transmitting/receiving optics of the reference system. This guarantees that the optical configuration of the reference system has not changed between the realignment and the retrieval of $O_{1}\left(z\right)$ and that the overlap function corresponds to the actual setup of the system.

Figure 1 shows the first of our measurements taken on 18 June 2015. Figure 1a shows the range-square corrected signal of System 2, $X_{2}^{ncor}$, as is, with no overlap correction, and the range-square corrected signal of System 1 scaled to that of System 2, $X_{1}^{cor}\cdot Norm$. As can be seen, good agreement is observed between the signals observed for z>4 km so that $z_{0}$ was set to 4 km. However, below 3–4 km, the narrow field-of-view System 2 signal shows the expected fall-off due to the influence of $O_{2}$. The ratio of these signals, $\frac{1}{Norm}\frac{X_{2}^{ncor}}{X_{1}^{cor}}$, is shown in Figure 1b in green (without filtering) and in red (with filtering), and it represents the overlap function of System 2. The associated error is shown as a shaded red area. At $z=z_{0}=4\;km$, $\Delta O_{2}$ is 0.02 (2% with respect to 1.0).

## 4. Discussion

In Figure 2, we plot several retrievals of $O_{2}$ performed approximately every 6 months between June 2015 and January 2017. Figure 2a shows the overlap functions and Figure 2b shows the absolute and relative errors associated to them. The height $z_{0}$ was adjusted case by case by looking at the range-square corrected signals (see Figure 1a) and is reported in Figure 2a. It varies between 3 and 6 km. Below 1.5 km, all overlap functions have a very similar shape. Most of the discrepancy is found between 1.5 and 5 km. The error $\Delta O_{2}$ due to the noise associated to $X_{2}^{ncor}$ and to the error associated to the retrieval of $X_{1}^{cor}$ is less than 0.07 at 6 km for all cases. In relative terms, $\Delta O_{2}/O_{2}$ at $z=z_{0}$ is 2.0, 1.2, 1.4, and 7.0% on 18 June 2015 (150618), 12 January 2016 (160112), 20 June 2016 (160620), and 14 January 2017 (170114), respectively. In all cases, the relative error stays within very reasonable values, in spite of the potentially large sources of error considered, as $\Delta \beta_{Raman}^{}$ and $\Delta \beta_{elastic}^{}$ may be (see Appendix A for description).

The temporal variation of the overlap function of System 2 is analyzed by comparing the retrievals of 160112, 160620, and 170114 to the one of 150618. Regarding the retrieval of the overlap on 150618, the first one in time is taken as a reference to which the following (in time) retrievals (on 160112, 160620, and 170114) are compared. This choice of the reference overlap is purely chronological. Absolute and relative differences are shown in Figure 3. The profiles on 160112 and 1600620 are steeper than on 150618 (positive absolute difference), while on 170114, the profile is slower (negative absolute difference). Again, below 1.5 km, differences are small: absolute differences are less than 0.06. Although the number of retrievals is not enough for the analysis to be statistically representative, this result is still a comforting one, since it shows that the overlap function variability in the near range remains small, at least over the 1.5-year period considered. Peaks in absolute difference are reached at 1.7 (160112 and 160620) and 2.4 km (170114). They are associated to peaks in relative terms of +18 and −13%, respectively. The positive difference of 160112 and 160620 vs. 150618 above 0.6–0.7 km suggests a “quicker” overlap function, i.e., that the transmitter and receiver optical axes on 160112 and 160620 are better aligned compared to 150618. Contrarily, the negative difference of 170114 vs. 150618 above 1.35 km and up to 5–6 km suggests a “slower” overlap function, i.e., that the transmitter and receiver optical axes on 170114 are more divergent than on 150618. Since the MPL system is not aligned on a regular basis, these changes in the alignment of the transmitting and receiving axes occur naturally and are due to thermal–mechanical changes in the system. The analytical retrieval of the overlap function of System 2 and its error bar are quite straightforward. That is why such calculations should be performed at all sites running an elastic lidar or a ceilometer whenever a reference, overlap-corrected system is available nearby.

Finally, we discuss the implication of the use of the different overlap functions obtained on the retrieval of the aerosol optical properties, namely here the backscatter coefficient, since the MPL system is an elastic system. To this end, we have performed elastic inversions [17] with a constant lidar ratio of 50 sr of the measurements made on 18 June 2015. The inversion was performed for System 1 (the reference system) corrected for its overlap and then for System 2 (the MPL) corrected for the four different overlaps obtained. The resulting profiles of the backscatter coefficient are presented in Figure 4 (left). The absolute and relative difference between the profiles corrected for the overlaps 160112, 160620, and 170114 instead of 150618 are also shown in Figure 4 (center and right). The vertically integrated backscatter coefficient multiplied by the lidar ratio employed, 50 sr, is an approximation of the aerosol optical depth (AOD), and it is reported in Figure 4 (left). The backscatter coefficients from the reference system and from the MPL with the overlaps 150618, 160112, and 160620 are similar. The only differences visible to the naked eye are observed at peaks of the profile at 1.7 and 2.0 km and in the first 500 m. The backscatter coefficient from the MPL with the overlap 170114 is notably different from the others above and below 1.7 km. To quantify the implication of using overlaps 160112, 160620, and 170114 instead of 150618, we now look at the middle and right plots of Figure 4. Below 0.5 km, the negative difference of 160112 and 160620 (Figure 3) results in an overestimation (as high as 50%) of the backscatter; the positive difference of 170114 (Figure 3) results in an underestimation (as high as 60%). Above 0.5 km and up to the main aerosol layer (≈3 km), the use of the overlaps 160112 and 160620 induces a backscatter underestimation as high as 19%. In the same height interval, the use of the overlap 170114 induces a relative difference varying from −60 to +50%. In terms of column-integrated magnitude, no significant difference is observed: the AODs agree within −0.001/+0.007 with the AOD from the reference system. This interesting result shows that the backscatter overestimation caused by a smaller overlap in the range (0; 0.5 km) is compensated by the underestimation in the above range (0.5; 3.0 km) due to a higher overlap. This is the case for 160112 and 160620. For 170114, the opposite occurs. The analysis of Figure 3 and Figure 4 together also demonstrates the non-linearity of the solution of the two-component elastic lidar inversion algorithm. One sees for example that although the overlaps on 160112 and 160620 are very similar, almost undistinguishable, below 1 km (Figure 3), the difference they produce on the backscatter retrieval is significant: e.g., at 0.3 km, the relative difference they produce is 5 and 33%, respectively.

## 5. Conclusions

This paper establishes simple mathematical relationships between range-square corrected signals from a lidar system corrected for the overlap effect and from another lidar system, or a ceilometer, not corrected for its overlap. The method allows the estimation of the overlap function of the second system as well as the associated error bars. The error is expressed as a function of the noise associated to the signals of both systems as well as the error of the overlap function of System 1. All error sources are given analytically either in the main body of the paper or in Appendix A. Several overlap functions have been retrieved with this method over a period of 1.5 years with two lidar systems of the Universitat Politècnica de Catalunya, Barcelona, Spain. System 2, i.e., the lidar system, the overlap function of which has been estimated is a micro pulse lidar system. The error associated to the estimated overlap is usually less than 7% when the overlap reaches unity. The temporal variability of profiles of overlap functions estimated over a period of 1.5 years is less than 11% in the first 1.5 km from the surface and peaks at 18% at heights between 1.7 and 2.4 km. The implication of these results to retrieved optical properties is also analyzed. The effect of using an inappropriate overlap function in the retrieval of the backscatter coefficient yield errors up to 60% in the first 0.5 km and up to 20% above. The sign of these errors is the opposite of the difference between overlaps. These high differences in backscatter when an inappropriate overlap is used highlight the need for regular overlap estimations, which can be easily made with the expressions given in this Letter whenever a reference, overlap-corrected system is available nearby.

## Figures and Tables

**Figure 1 sensors-20-06312-f001:**
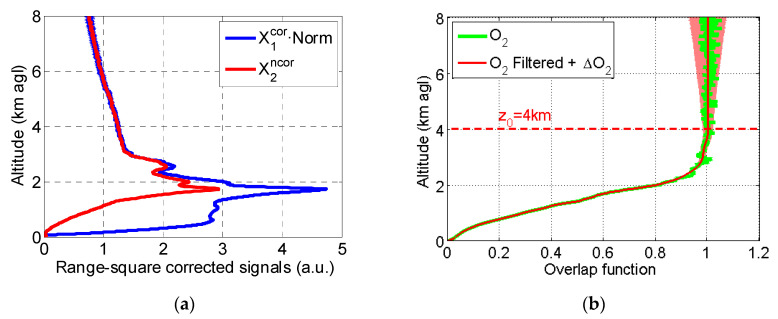
(**a**) Range-square corrected signals of System 1 and System 2; (**b**) Overlap function of System 2. The shaded red area represents $\Delta O_{2}(z)$. The measurement was performed on 18 June 2015 (150618).

**Figure 2 sensors-20-06312-f002:**
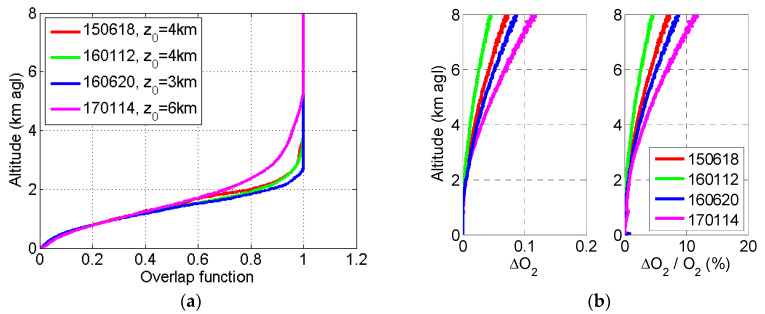
(**a**) Several retrievals of the overlap function of System 2; (**b**) Absolute $\left(\Delta O_{2}\right)$ and relative $\left(\Delta O_{2}/O_{2}\right)$ errors associated to these retrievals.

**Figure 3 sensors-20-06312-f003:**
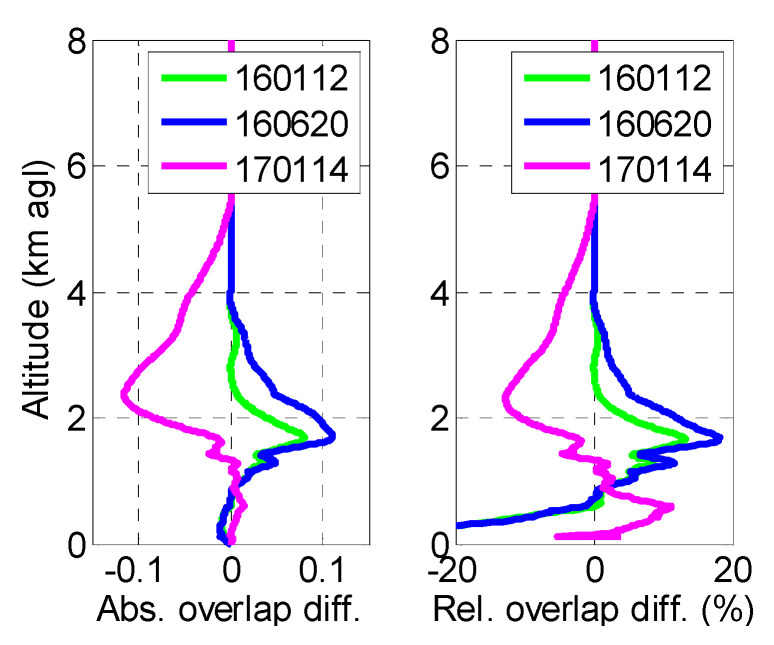
(**Left**) Absolute and (**right**) relative difference between the overlaps retrieved on 12 January 2016, 20 June 2016, and 14 January 2017 vs. 18 June 2015.

**Figure 4 sensors-20-06312-f004:**
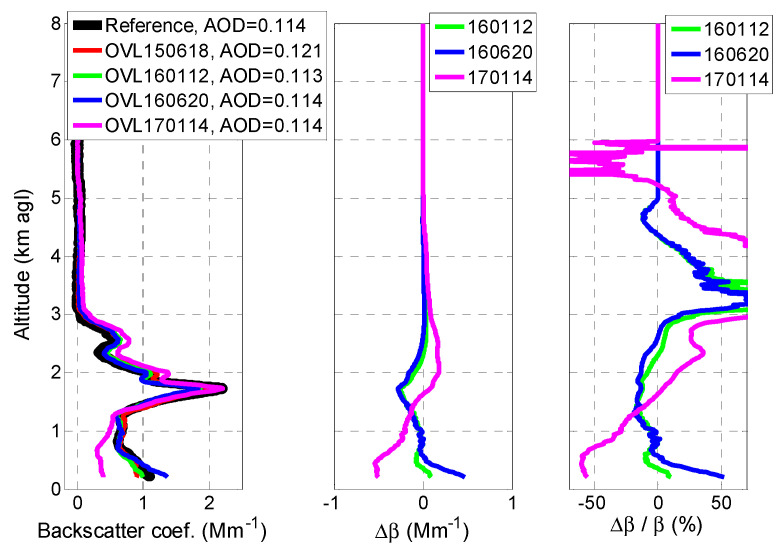
(**Left**) Backscatter coefficient on 18/6/2015, (**center**) absolute difference in backscatter when using the overlap 160112, 160620, and 170114 instead of 150618 and (**right**) relative difference in backscatter when using the overlap 160112, 160620, and 170114 instead of 150618.

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
