# Peer review of "Calculation of the Overlap Function and Associated Error of an Elastic Lidar or a Ceilometer: Cross-Comparison with a Cooperative Overlap-Corrected System"

_sensors, 2020, doi:10.3390/s20216312_

Round 1

Reviewer 1 Report

Summary: This paper describes a method for estimating the overlap function and associated error of an elastic lidar system when another system with a known overlap function is available. Several overlap functions were retrieved using 2 lidar systems and the overlap and error were evaluated. Errors observed in the overlap function were <18%.

Major Comments:

My primary issue with this manuscript is that critical descriptions are lacking. I specifically use lidar systems for evaluating ocean processes/biological constituents, so found this paper very hard to follow. I found that reading a few of the references cited in this paper, for example [12], gave me a much better/clearer description of the issues associated with calculating the overlap function. If the reader isn’t familiar with the lidar technique for atmospheric applications and associated measurement issues, this paper isn't complete on its own.

Additionally, the equations don’t seem to flow/follow a single train of thought. For example, moving from Eq. 1 to Eq. 2, the authors apparently switch from using beta, C, and transmission in Eq. 1 to received Power in Eq. 2. Maybe I am misunderstanding something; again, my specific field of research applies different equations for solving for lidar-retrieved ocean properties. However, reading [12] and other sources, I was more easily able to follow the mathematical reasoning. Perhaps moving the Appendix to the main body would help with this issue.

My last major issue is that the Discussion section reads more like Results. The authors don’t really discuss how their results apply for retrieval of aerosols/pollutants.

Minor Comments:

Ln 18 “to estimate” should be “the estimation of”

Ln 20 I think year should be plural

Ln 23-24 This comment relates to the first major comment above. I think this sentence could be more descriptive; the wording “first 1.5km” was unclear to me.

Ln 32-35 When using a semicolon, full sentences are separated.

Ln 39, 40, 69 When citing an example, I don’t think you need to use e.g.

Ln 110 “since” should be “by” I think.

Author Response

Point-to-point answers are given in the attached PDF. Thank you.

Reviewer 2 Report

The manuscript proposes a new calculations approach of the overlap function.

I have some major concerns regarding this manuscript because there are several drawbacks. One of them is scientific writing [e.g. “That is why we encourage….”, “to cite but two examples”– it is about a scientific paper in the physical sciences and engineering field, not a conference or workshop; please reformulate]. Next, the manner of presentation diminishes the potential importance of the study. This paper shows no conclusions!

The goal looks quite ambitious and worth investigating, however from the manuscript it is not clear which are the strong points.

I kindly ask the authors to include more lidar characteristics and to develop the discussions about the negative absolute difference for the 170114 profile. Since number of retrievals is not enough for the analysis to be statistically representative, I kindly ask the author to discuss why their considered only the 150618 profile as reference [this can create controversial misunderstandings] and better motivate and discuss about the importance of the system 1 in estimating the overlap function of System 2.

If the author prefers to keep this paper as a letter, please remove the appendix to improve this study and keep only the references where appropriate in text.

Minor aspects:

- line 166: please move up the round bracket

-  figure 3: “(a)”, makes no sense

- The authors write that Supplementary Materials are available online at www.mdpi.com/xxx/s1, Figure S1: title, Table S1: title, Video S1: title. I could not find them.

- line 112: “real” – please delete this word and reformulate. It goes without saying that the systems are real, specially that the lidar systems are part from the recognized lidar networks.

- I kindly ask the authors to reformulate some phrases that repeat some aspects.

Author Response

(The authors gave the same response as above.)

Reviewer 3 Report

The article describes the relationship between the signals of 2 lidar systems - one with corrected overlap effect and the other unknown. The technique presented in this paper is interesting from a scientific point of view. However, analysis boils down to precisely knowing the ratio of the system constants of the lidar systems being considered. Below are my comments:

  1. The system constant of a lidar system changes in time. This implies that the Norm also changes. How good is the method when applied to, say 24 hour operation? Is the the variation of the norm small enough and by how much?
  2. It is not clear how or what the authors criteria to choose the data to implement this method. Are the data under the same atmospheric conditions?
  3. Line 120-121: How about the application of this analysis on daytime cases? Why only focus on nighttime?
  4. Line 132: It looks like the two signals are the same at 3 km. What is the criteria used by the authors to say that there is a good agreement of the two signals at 4km?
  5. By knowing the overlap correction, the authors have not discussed the implication of this results to retrieved optical parameters from lidar signals, e.g., extinction coefficient.
  6. Lines 154-156: It will be good if the author can discuss the differences of the profiles.

Author Response

(The authors gave the same response as above.)

Round 2

Reviewer 2 Report

The authors have solved all the problems raised by me and this letter type can be published after minor revisions mentioned below.

- please correct the formula 10 due to some errors to pdf converting, I suppose.

- Line 126: please put “2” as a subscript or write “nitrogen”

On the other hand, in my opinion is that a letter type work should not be accompanied by an appendix, especially since it is two known formulas that can be easily included in the text and cited, and readers will find them even more useful (not all are specialist in lidar systems). But I will leave it to the editors' decision on this formal issue.

Author Response

Thank you for the final review. Some answers:

  • Eq. (10) is fine in the Word document submitted by the authors. However it is true that in the pdf generated by MDPI editorial office a chinese symbol appears in the middle of the equation. We will double check the peer-reviewed document to make sure Eq. (10) is fine in the published article.
  • Line 126: "2" has been put in subscript.
  • Appendix: the editor did not say anything about this issue, so the authors decide to keep Appendix A.
